# Structural Health Monitoring of Walking Dragline Excavator Using Acoustic Emission

**Vera Barat** [1,2], **Artem Marchenkov** [1,*], **Dmitry Kritskiy** [3], **Vladimir Bardakov** [1,2], **Marina Karpova** [1], **Mikhail Kuznetsov** [1], **Anastasia Zaprudnova** [1], **Sergey Ushanov** [1,2] and **Sergey Elizarov** [2]

1   Moscow Power Engineering Institute, 14, Krasnokazarmennaya Str., 111250 Moscow, Russia; vera.barat@mail.ru (V.B.); bardakovvv@interunis-it.ru (V.B.); karpova.m.v24@gmail.com (M.K.); mikx10@mail.ru (M.K.); morningcoffee@inbox.ru (A.Z.); ycergey@mail.ru (S.U.)
2   LLC "Interunis-IT", 20b, Entuziastov Sh., 111024 Moscow, Russia; serg@interunis-it.ru
3   JSC "SUEK": 20b, Lenin Str., 660049 Krasnoyarsk, Russia; kritskijdy@suek.ru
*   Correspondence: art-marchenkov@yandex.ru

**Abstract:** The article is devoted to the organization of the structural health monitoring of a walking dragline excavator using the acoustic emission (AE) method. Since the dragline excavator under study is a large and noisy industrial facility, preliminary prospecting researches were carried out to conduct effective control by the AE method, including the study of AE sources, AE waveguide, and noise parameters analysis. In addition, AE filtering methods were improved. It is shown that application of the developed filtering algorithms allows to detect AE impulses from cracks and defects against a background noise exceeding the useful signal in amplitude and intensity. Using the proposed solutions in the monitoring of a real dragline excavator during its operation made it possible to identify a crack in one of its elements (weld joint in a dragline back leg).

**Keywords:** acoustic emission; structure health monitoring; walking excavator; AE impulse detection





## 1. Introduction

The dragline excavator is a widely known type of heavy equipment used in civil engineering and surface mining industry. Commonly, a system of a dragline consists of a large bucket that is suspended from a boom with wire ropes (Figure 1). The hoist rope is usually driven by diesel or electric motors. Its main function is to support the dragline bucket and hoist-coupler assembly from the boom. The dragrope is generally applied to move the bucket horizontally. Unlike conventional types of excavators, dragline digs by dragging the bucket only in one direction—towards the excavator [1].

Currently, most dragline systems are operated in the open pit mining industry to process blasted rocks located both above and below the horizon level of the dragline excavator. Moreover, draglines have large dimensions—the boom length may reach 90 m, the total height can constitute up to 100 m, and the bucket capacity may achieve 20 tons. Nowadays, the design of draglines is favorable due to a short working cycle and high performance. Currently, more than 500 walking dragline excavators are in operation in the world, of which more than 200 are in Russia. Leading dragline manufacturers are Caterpillar, Liebherr, Uralmash, Bucyrus, and others.

Draglines operate all year-round in detrimental weather conditions: significant temperature fluctuations, great levels of humidity, and even regions with increased seismic activity. Both the influence of external weather conditions and constant cyclic loadings can lead to the formation of fatigue fracture-initiating defects in the metal elements of the dragline structure. Such defects can reduce the operation life of the equipment due to premature destruction. The breakdown basically causes considerable financial costs associated with equipment downtime, repair costs, and terrible accidents [2]. The most common type of dragline excavator damage is cracks in the back legs; for the period from

2015 to 2020, 15 accidents caused by breakage of back legs and five accidents caused by breakage of the dragline boom were registered in Russia.

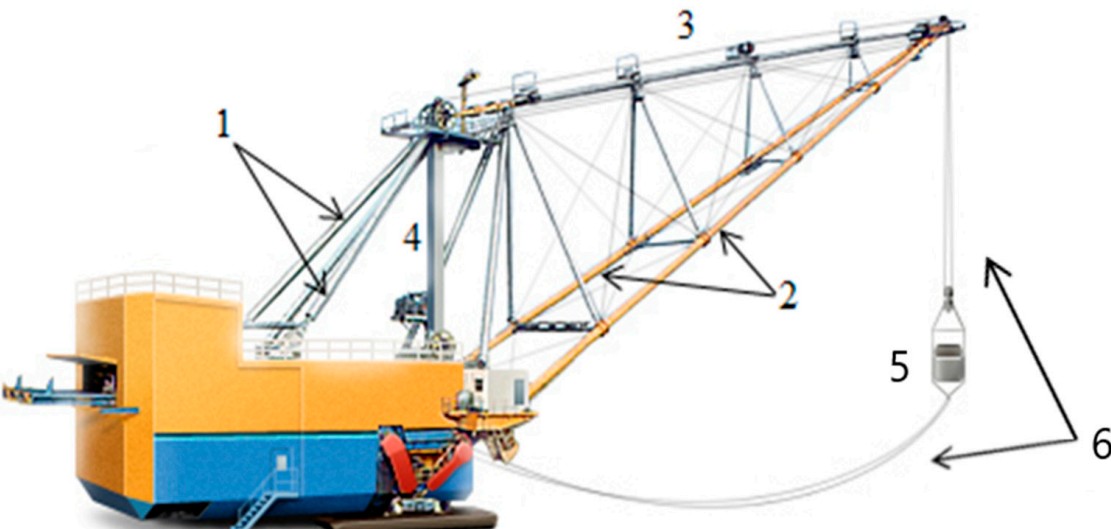

**Figure 1.** Basic components of a dragline ESH-20.90: 1—back legs, 2—boom, 3—hoist rope, 4—mast, 5—bucket, 6—dragrope.

The most widely spread non-destructive testing (NDT) of dragline structural components is ultrasonic testing, which is carried out in the welded joints of an excavator twice a year. However, this method does not allow detect abrupt changes in the stress–strain state. Furthermore, it is extremely time-consuming and leads to long-lasting downtime. Draglines are commonly equipped with a structural health monitoring system with video supervision systems and an electrical equipment diagnostics system, but operational monitoring of the metal structures conditions is usually not performed.

In this paper, the possibility of using the acoustic emission (AE) method to detect defects in the dragline metal structures is discussed.

AE is a non-destructive testing (NDT) method based on the physical phenomenon of elastic waves emission when a material undergoes irreversible changes in its internal structure, for example, because of crack formation. The AE method has a high sensitivity to crack detection and is used to detect cracks at early stages of their growth. An important advantage of the AE method is the ability to detect defects located at a great distance from the control point; cracks and other defects emit AE waves, which propagate over the testing structure and can be detected by the AE sensors at about 10 m from the defect. The AE method is passive, it does not require probing action, which allows it to be effectively used in structural health monitoring systems for monitoring loaded structures in the operating mode without decommissioning.

AE testing provides low noise immunity, therefore, such experiments are traditionally carried out for equipment when the operation of equipment is suspended. Recently, the technology of AE monitoring which requires the use of specific methods for detecting AE signals against a noise background has become more popular [3–5]. AE monitoring is performed for both statically and dynamically loaded equipment. Most widespread application of AE testing in the operating mode is testing of bearings, compressors, and pumps [6–8]. The various defects have a specific AE signature with a certain periodicity connected with a rotation frequency. The periodicity analysis allows to recognize the AE activity against the noise process. Some of the most complicated structures for AE structural health monitoring are railways and roads bridges [9,10]. The complicity consists of the fact that trains and cars are oftentimes a noise source and a load excitation, and the moment of passage of vehicles is simultaneously the moment of the most probable occurrence of AE activity. Solving of this problem is achieved by reducing the distance between AE sensors,

high-frequency filtration, and using additional strain gauges or vibration measurements. Another way to detect the AE activity of a defect during the testing of the noisy structure is to design a scaled model of the testing structure. This method is successfully used for the AE testing of the various chemical reactors, absorbers, and crystallizers [11,12]. Investigation of the scaled model of the test structure allows obtaining a pattern of acoustic noise, in this case the appearance of a defect can be detected as a change of regularity in the parameters of the reference noise signal. If it is impossible to design a model, the study of noise is carried out on the operating equipment. There are some papers devoted to the AE testing of handling equipment: cranes, lifts, and excavators [13–17]. However, as a rule, the testing of handling equipment involves a special low-speed loading program which provides a low noisy level. AE testing of a walking dragline excavator is a rather difficult problem, since it involves the testing of a large-sized structure in difficult weather conditions and at an extremely high noise level.

This paper describes a method of design of a walking dragline excavator monitoring system based on the application of the AE method. A simplified block diagram of the monitoring system is shown in Figure 2. It consists of hardware part and software parts, the software parts in turn are divided into acquisition system and data analysis system.

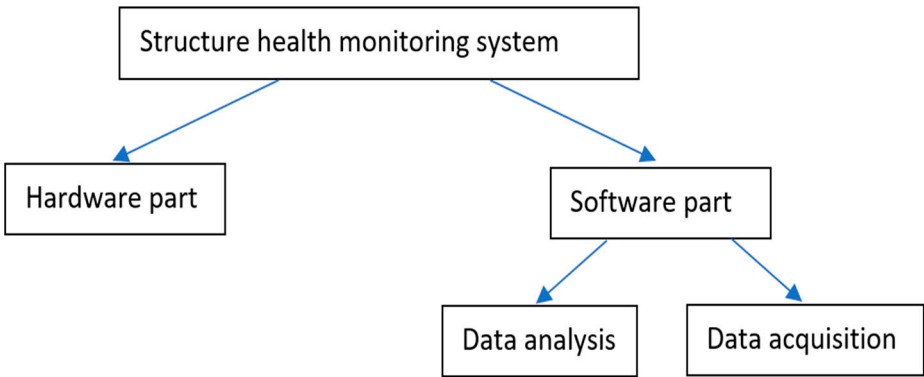

**Figure 2.** Block diagram of the structure health monitoring system.

The hardware equipment is represented by a digital measuring system A-Line 32D DDM (produced by Interunis-IT company). Its main feature consists of subsequential connection of the measuring channels. A data acquisition system provides the traditional AE impulse threshold detection, calculation of AE parameters, and location of AE sources based on the determination of the difference in the time of arrival of AE impulses to neighboring sensors. The data analysis algorithm provides intelligent processing of AE data for detection of the defects against the background of high-intensity technological noise.

In a preliminary analysis, the possibility of carrying out AE monitoring to assess the strain–stress state of the dragline components was studied by identification of favorable and adverse factors. The most unfavorable factor is a high level of noise in the dragline operating cycle. In most cases it is caused by the movement of the dragrope and hoisting ropes. Therefore, the high-intensity acoustic noise can reach 75 dB in the 100–400 kHz range. Despite this critical level, monitoring still can be performed because the sources of noise, as a rule, are not localized at certain points in the structure and are not identified on the diagram. The favorable factor is the waveguide form. The dragline structural components usually obtain the form of hollow pipelines and could be interpreted as a one-layered cylindric waveguide. In the absence of multiple reflections and scattering, the AE impulses have a relatively short rise time and duration, while providing a significant difference in the waveform of the useful and noise data.

## 2. Materials and Methods

This section represents methods for analyzing various factors which impact on the AE structural health monitoring technology and collecting information about the AE source, waveguide properties, and the features of the structure loading in the operating cycle, as well as the parameters of noise.

### 2.1. AE Source Characterization

Mechanical tests were carried out on specimens made of the same steel as the dragline back legs to determine the AE parameters of a fatigue crack. Specimens were produced from 16G low-alloyed steel (marked in accordance with Russian state standard) sheets using laser cutting. The chemical composition of the studied steels is presented in Table 1.

**Table 1.** The chemical composition of the 16G low-alloyed steel.

| The Content of Chemical Elements (% wt.) | | | | | | | | |
|---|---|---|---|---|---|---|---|---|
| C | Mn | Si | Cr | Ni | Al | Mo | S | P |
| 0.14–0.18 | 1.0–1.2 | 0.15–0.3 | ≤0.2 | ≤0.3 | 0.02–0.05 | ≤0.8 | ≤0.02 | ≤0.025 |

The shape and overall dimensions of the specimens (Figure 3a) were selected on the basis of Russian state standard GOST 25.506-85. Thus, a series of rectangular specimens with a thickness of 5 mm and with an edge notch were prepared. The width is equal to $b = 50$ mm, and the length $l = 350$ mm was chosen due to the condition of $l \geq 2b$ and taking into account the distance between the grips. The notch width $e$ also should meet the requirement ($e \leq 0.06b$) and was selected as $e = 4$ mm, the opening angle $\Theta$ corresponded to ($30° \leq \Theta \leq 60°$) and was $\Theta = 45°$, and the notch depth $h = 10$ mm.

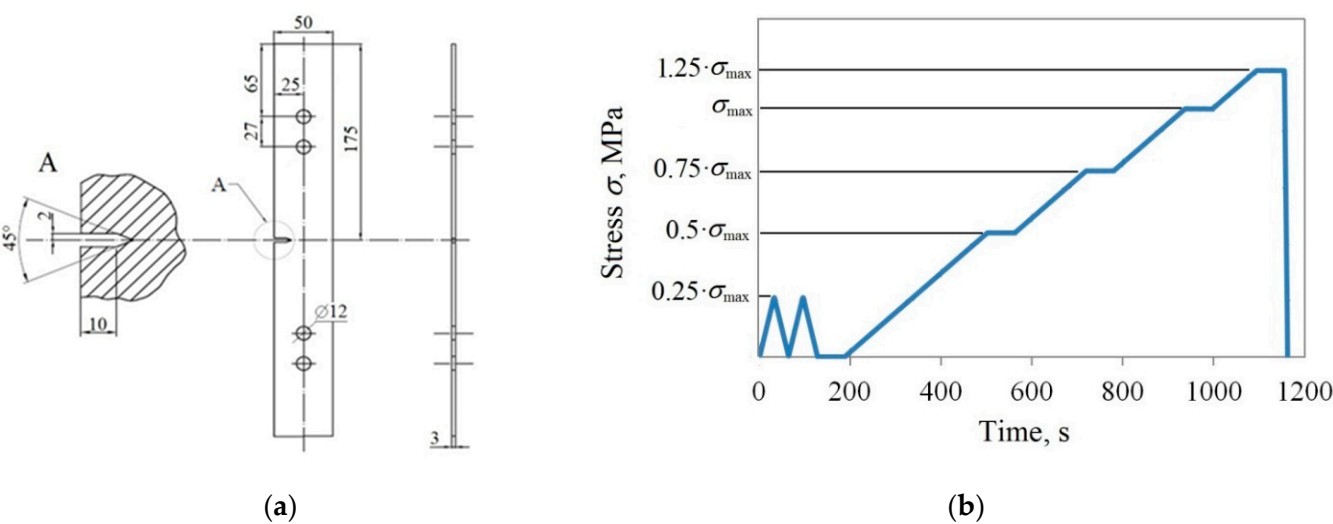

(a)                    (b)

**Figure 3.** The design of the test specimen: (**a**) scheme of the specimen; (**b**) diagram of the loading cycle.

A fatigue crack was grown on each specimen by cyclic tensile loading using the pulsation cycle (cycle asymmetry coefficient $R_c = 0$) with a frequency of 5 Hz and a maximum cycle load of $\sigma_{max} \approx 0.6 \cdot \sigma_y$, where $\sigma_y$ is a yield stress. The number of loading cycles was chosen so that the fatigue crack developing under cyclic loading in the lateral notch region would reach a certain length. Nominal crack lengths ranged from 3 to 15 mm. Thus, the total length of the notch and crack was approximately 15–28 mm depending on the specimen. After the crack growth, the specimens were tested under static tension using a loading scheme adopted during industrial AE testing (Figure 3b) [18]. A series of experiments was done for 10 specimens with identical parameters.

The A-Line 32D AE system (LLC "Interunis-IT") was chosen during the experiment. It was equipped with four measuring channels including the PAEF-014 preamplifier and GT200 resonant sensor (LLC "GlobalTest", resonant frequency 180 kHz). The intrinsic noise of the equipment, preamplifier, and AE sensor was 26 dB. The discrimination threshold was selected as 45 dB (6 dB higher than the noise level of the testing machine).

Figure 4 presents the dependence of the cumulative AE hits depending on the current stress and time. Cumulative AE hits have a power-law dependence of the $\sigma/\sigma_y$ relation which is typical for the crack presence. The number of AE hits registered during the loading of the specimens varied in the range from 118 to 240 with an average value about of $N_\Sigma = 185$. The number of AE hits depends on the loading stress. For more objective evaluation of the material emissivity the Palmer–Heald model was implemented.

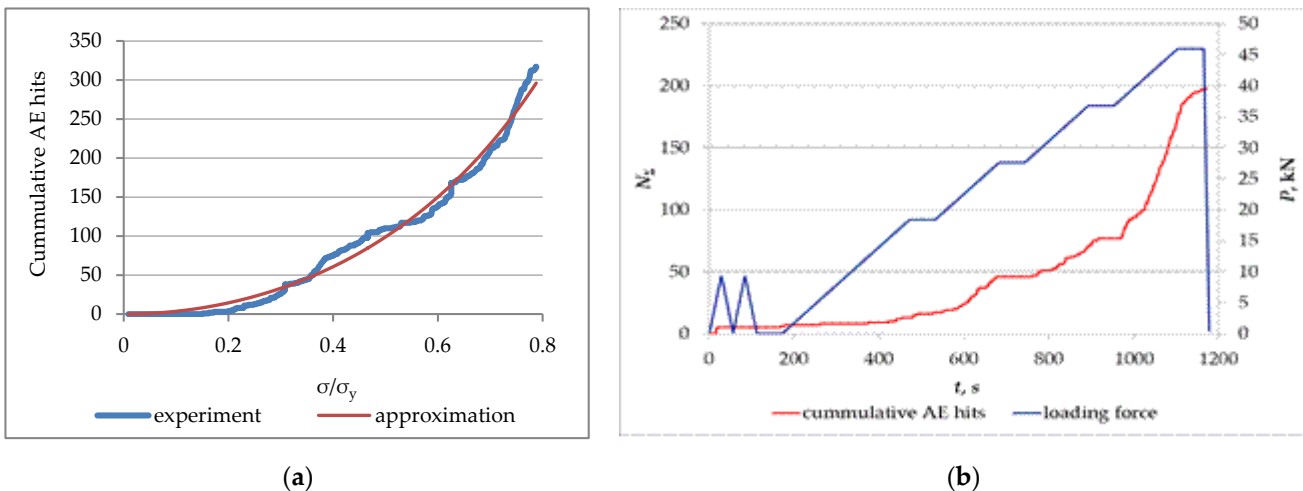

(a)                                                (b)

**Figure 4.** Cumulative acoustic emission (AE) hits $N_\Sigma$ relations: (**a**) cumulative AE hits $N_\Sigma$ vs. current stress $\sigma/\sigma_y$ and its Palmer–Heald approximation; (**b**) cumulative AE hits $N_\Sigma$ and loading force $P$ vs. time $t$.

According to the Palmer–Heald model,

$$N_\Sigma \; = \; D{\cdot}a{\cdot}\left(sec\left(\frac{\pi}{2}\frac{\sigma}{\sigma_y}\right) \; - \; 1\right) \tag{1}$$

where $\sigma$ is the actual stress; $\sigma_y$—yield stress; $a$ is half the length of the crack; $D$—dimensional coefficient of the model (1/mm), depending on the characteristics of the material, temperature, and type of stress–strain state.

It is presented in [19] that the multiplicative coefficient of the Palmer–Heald model $D$ can be considered as parameters that may be applied to describe the material AE emissivity independent of loading conditions, crack length, and stress intensity factor. Dependence of cumulative AE hits vs. time was approximated using the Palmer–Heald equation for the data obtained after testing of each 10 specimens in the series. An example of such an approximation is given in Figure 4b. For each dependence, the values of the parameter $D$, the error $\delta D$, and the coefficient of determination $R^2$ were determined and presented in the Table 2. The parameter $D$ was calculated based on the values presented in Table 2 and constituted 151.1 ± 8.4 (1/mm).

**Table 2.** AE emissivity estimation—the Palmer–Heald model parameters.

| $D$, 1/mm | $\delta D$, % | $R^2$ |
|---|---|---|
| 137.5 | 0.93 | 0.97 |
| 149.6 | 0.87 | 0.96 |
| 144.9 | 1.4 | 0.92 |
| 152.3 | 0.82 | 0.98 |
| 151.4 | 0.93 | 0.96 |
| 155.3 | 1.4 | 0.99 |
| 165.1 | 3.1 | 0.95 |
| 160.5 | 1.5 | 0.94 |
| 153.2 | 1.3 | 0.98 |
| 141.3 | 2.1 | 0.97 |
| 144.9 | 1.4 | 0.92 |

Since the maximum load in the operation cycle of the dragline is approximately $0.6\sigma_y$, the approximately number of AE hits related to crack length is $N_\Sigma/a = D\ (\sec(0.3\pi) - 1) \approx 106$.

Figure 5 shows the dependence of the amplitudes of AE impulses emitted by the crack depending on time (Figure 5a) and the AE impulse amplitudes probability distribution in double logarithmic axes (Figure 5b). The amplitudes of the AE impulses emitted by the crack do not exceed 72 dB, and the amplitude distribution corresponds to the Weibull law with the parameters $k = 1.17$, $\lambda = 6.13$.

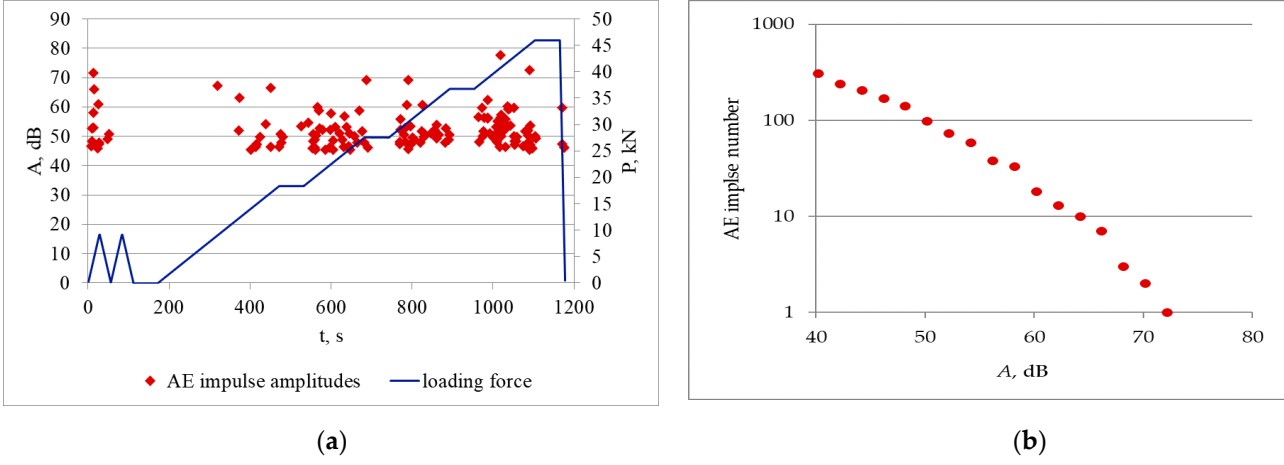

(**a**)           (**b**)

**Figure 5.** Amplitudes of AE impulses: (**a**) AE impulse amplitudes $A$ and loading force $P$ vs. time $t$; (**b**) AE impulse amplitudes distribution.

Assessment of the AE emissivity and determination of the range of AE impulse amplitudes emitted by the crack are used to determine the parameters of the AE structural health monitoring system and estimate the reliability of AE impulse detection against background noise.

### 2.2. AE Waveguide Characterization

In the previous section, the source of AE is characterized. However, to detect AE signal impulses effectively against a background of noise, it is necessary to assess how AE parameters vary during propagation along the waveguide. Traditionally, the investigation of the waveguide of the testing objects is carried out mainly empirically by measuring the Hsu-Nielsen lead break response at various points of the structure. In this study an analytical approach is implemented due to limited access to the dragline constructive elements. Load-bearing metal structures of dragline such as back legs and booms represent hollow tubes with a diameter of 20 inches which could be considered as a single-layer cylindrical waveguide. Ordinary form of the waveguide allows to calculate AE signal propagation analytically using modal analysis [20].

Modal analysis is a common method for the waveguide calculating based on the properties of orthogonality and completeness of normal waves. It allows to calculate in Fourier space the displacement of the testing structure at the point $z$ caused by the action of a point source with the waveform $u(0,t)$. The result signal is determined using the expression

$$u(z, f) = u(0, f) \cdot a_p(z, f) \qquad (2)$$

where $u(0,t)$ is the signal emitted by the AE source located in the point $z = 0$, and $u(0,f) = F[u(0,t)]$ is its Fourier transform. It was shown in [21] that when the external force is a point source, the coefficients $a_p(z)$ can be determined using the simplified formula:

$$a_p(z) = e^{jk_p z} / 4 \qquad (3)$$

where $k_p = (2\pi f)/c_{ph}(f)$ is the wave number; $c_{ph}(f)$ is the phase velocity of the normal wave, $f$—frequency.

In the case of a cylindrical waveguide with a large radius of curvature, Lamb waves can be considered as the main type of normal waves.

One of the main aspects related to the influence of the waveguide is attenuation. The attenuation effect is a difficult problem from the point of view of interpreting the AE data, since different wave modes and different frequency components are characterized by different attenuation coefficients, which has a significant effect on the waveform of AE impulse. The attenuation coefficients of the Lamb waves modes $\gamma_{S0}$ and $\gamma_{A0}$ are a linear combination of the attenuation coefficients of the longitudinal wave $\alpha$ and transverse wave $\beta$:

$$\begin{cases} \gamma_{S0} = A_{S0}\alpha + B_{S0}\beta \\ \gamma_{A0} = A_{A0}\alpha + B_{A0}\beta \end{cases} \qquad (4)$$

where $A_{S0}$, $B_{S0}$, $A_{A0}$, $B_{A0}$—coefficients that are determined by the type of Lamb wave, relative plate thickness, and Poisson's ratio.

Analytical expressions of the coefficients are given in [22], their values depend on frequency $f$ and thickness of the testing structure $h$. The analytical method based on the modal analysis, supplemented by analytical consideration of the attenuation coefficient and the impulse response of AE sensor, allows simulating AE signals of a realistic form [23]. Figure 6 represents the calculated signal (Figure 6a) and experimental signal obtained using the Hsu-Nielsen simulator (Figure 6b) for wall thickness 20 mm at 4 m from the AE sensor.

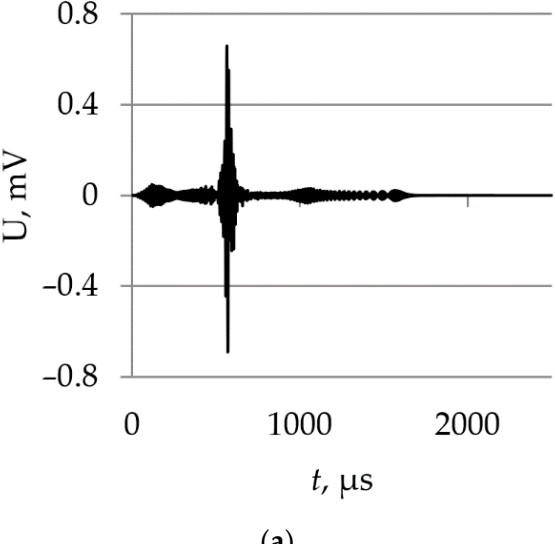

(a)

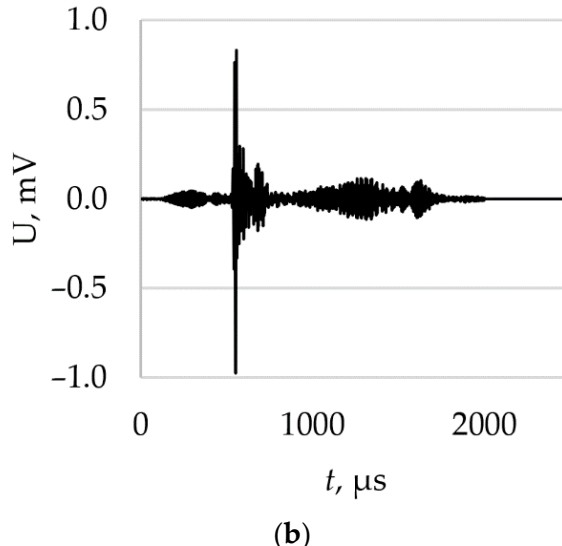

(b)

**Figure 6.** Comparison of the modelled and experimental AE signals *U* for a waveguide length of 4 m and a thickness of 20 mm: (**a**) the simulated signal, (**b**) the measured signal.

Based on the analytical method, a parametric model of the AE impulse can be designed. It presents the dependence of the AE parameters (rise time (*RT*), rise angle (*RA*), and duration) on propagation distance. The *RA* parameter represents the value inverse to slow rate and can be calculated with the equation

$$RA = RT/A, \tag{5}$$

where *A* is the amplitude of the AE impulse.

In the case of one-layered metallic waveguide, the AE impulse characterizes, as a rule, with some specific parameters, values such as a small *RA* and a short *RT*. The determination of the expected priory parameters of the AE impulse in advance makes it possible to identify them effectively against the background of noises, whose impulse components correspond to a larger time scale. Figure 7 shows *RT* and *RA* versus propagation distance *L*, which determine the expected values of AE parameters for a given distance between the AE source and sensor. These dependences can be interpreted as a parametric model of the AE impulse, based on which the identification of AE impulses against the background of noise will be carried out.

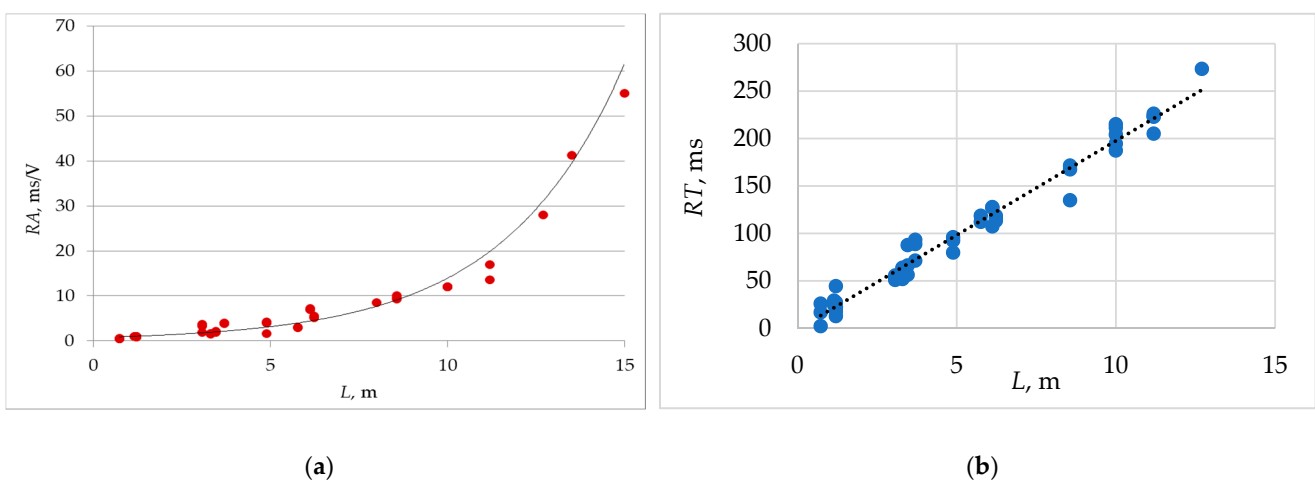

(**a**)　　　　　　　　　　　　　　　　　　　　　　　　　　　　　(**b**)

**Figure 7.** Dependence of AE parameters versus propagation distance *L*: (**a**) rise angle (*RA*) vs. distance *L*, (**b**) rise time (*RT*) vs. distance *L*.

*RA* varies exponentially with the distance, while *RT* increases linearly. Both parameters have rather low values; at 15 m, the *RA* does not exceed 50 ms/V, when the *RT* of the leading edge of the AE signal pulse for the same distance turns out to be less than 300 μs.

*2.3. Noise Parameters*

The main source of acoustic noise measured during the AE testing of walking dragline excavator is friction and vibration generated by the movement of the hoist ropes and dragropes that control the bucket position. The noise has a stochastic non-stationary nature, its intensity turned out to be different for AE sensors installed on various dragline constructive parts. The lower the noise intensity, the greater is the distance of the structural component from the moving ropes. Figure 8 shows the empirical probability function of the noise long-time realization measured with a different sensor located on the structural components above and below the boom, and on the side frame—back legs and a mast. The highest noise level was observed along the measuring channel installed in the lower part of the mast near the dragropes, while the lowest level was detected along the channels installed on the lower part of the boom. It can be explained by the fact that they are placed far away from the trajectory of the ropes.

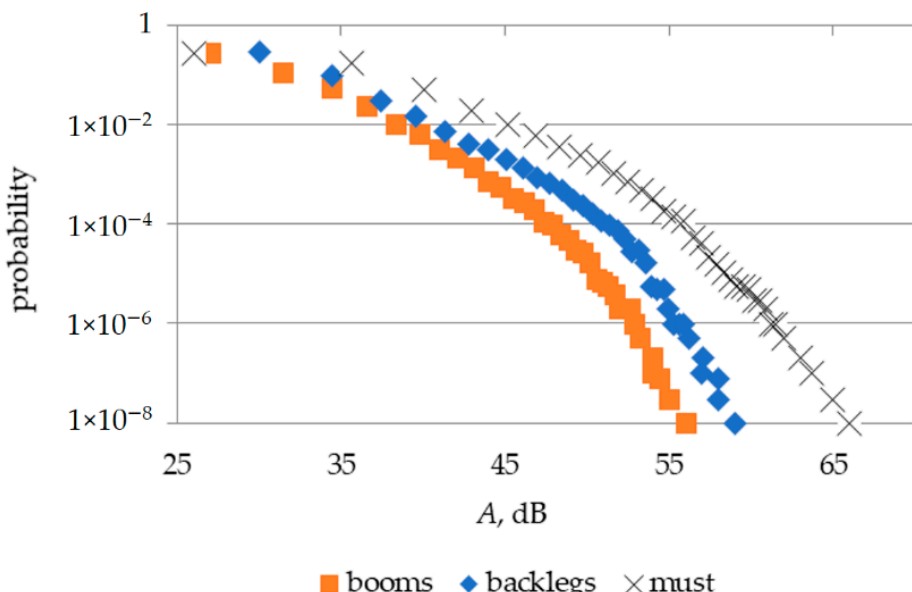

**Figure 8.** Empirical probability distribution of acoustic noise for a various dragline component.

It can be inferred from Figure 8 that there is a high level of noise with a standard deviation of 40–45 dB. The maximum values typical for a set of $10^8$ samples (that correspond to 100 s with a sampling frequency equal to 1 MHz) are 50–65 dB. The noise is non-stationary because it is caused by the uneven movement of the ropes in different phases of the dragline operation cycle. For instance, when the bucket is loaded, the dragrope is pulled. When there is a stage of unloading, both the dragrope and hoist rope are weakened and moved. When the platform turns, no movement of the ropes is visible, and the intensity of noise is dramatically reduced. Figure 9a presents the longtime noise waveform; its peak corresponds to 68 dB, and in the low-intensity phase ~ 40–45 dB. A favorable factor is that the impulse noise components differ from the AE signal impulses by greater values of the *RT* and *RA* parameters. Figure 9b shows the empirical distribution of the *RA* parameter corresponding to noise impulse components. The distribution mode equaled the value *RA* = 900 ms/V, which is significantly higher than the values of the *RA* parameter for AE impulses emitted by a defect.

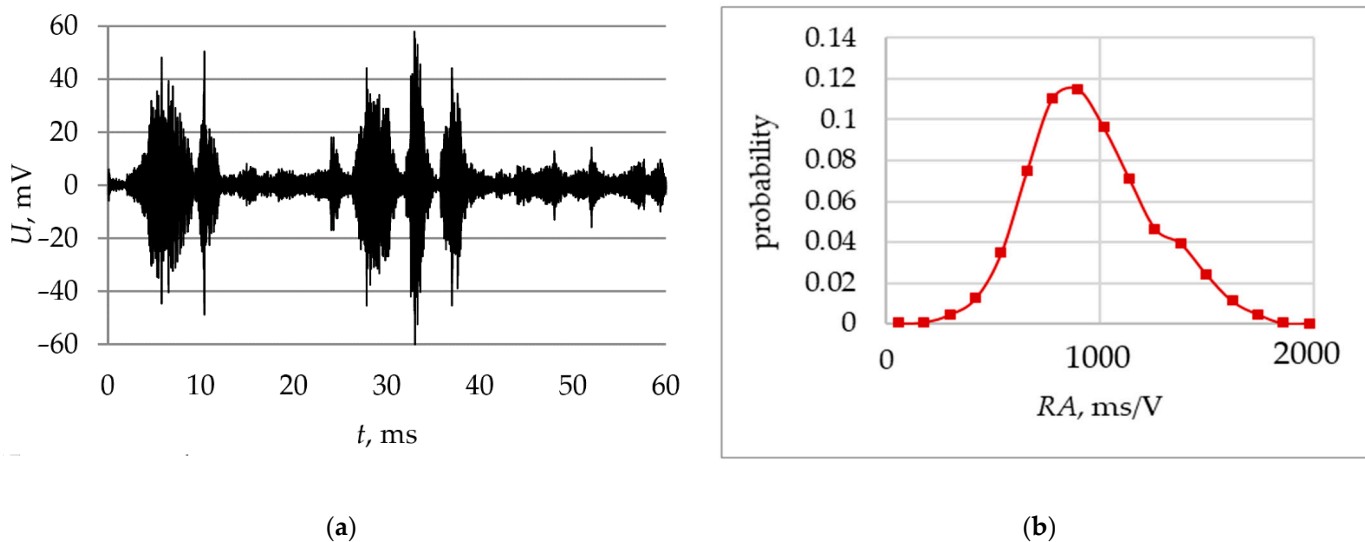

(**a**)             (**b**)

**Figure 9.** Parameters of the noise process: (**a**) noise waveform longtime realization, (**b**) distribution of *RA* parameter for the noise impulse components.

### 2.4. Filtering Method

Conventionally, the detection of AE impulses is carried out by the threshold method, in accordance with AE, impulses are detected during the threshold crossing. This method is highly effective and convenient in a case when a noise has a stationary character and can be rejected with the nonlinear threshold filtering. However, in the case of nonstationary impulse noise, the use of the threshold method is not acceptable. According to the AE non-destructive testing method, the threshold is set in such a way that no more than one false hit detection is recorded in 100 s. According to the probability distribution of noise samples (Figure 8), the amplitude discrimination threshold should be set between 55 and 65 dB for different locations. Such an increase in the amplitude discrimination threshold leads to a significant increase in the probability of missing a defect. A preliminary analysis of the AE source shows that at a value of the amplitude discrimination threshold of 45 dB, the average emissivity of the material is about 105 AE hits per 1 mm of crack growth. However, it can be extrapolated that with an increase in the discrimination threshold from 45 to 60 dB, the number of AE hits drastically decreases by about 94% due to the exponential distribution of the AE pulse amplitudes (Figure 4b). The emissivity in this case will be on the order of 5–7 AE hits per 1 mm of crack growth.

A special method was developed to increase the accuracy of crack detection in the dragline load-bearing metal structures. In accordance with this method, the amplitude discrimination level is set below the noise level to allow recording both the AE process and the noise technological process during the dragline operation. In this case, a type II error associated with the defect missing is minimized by increasing the probability of a type I error connected with false detection of a defect due to erroneous noise impulse location. Since the noise intensity is significantly higher than the AE hits count rate, special filtering is developed to increase the reliability of the testing. Filtering is effective and specific since it is based on the parametric model of AE impulse, representing a set of characteristic values of the AE parameters corresponding to a certain distance between the AE sensor and the defect.

A spatial parametric filter is proposed for detecting AE impulses from nonstationary impulse noise. This filter excludes from the location schemes those AE events whose parameters are more consistent with the noise process and are not typical for AE impulses caused by a defect. Since the tested elements of the dragline are long pipelines (the length significantly exceeds the diameter), a linear circuit with an antenna formed by two sensors is used to determine the location of the defect.

The filtering algorithm assumes that the waveform of AE impulse is determined primarily by the parameters of the acoustic waveguide. The acoustic emission impulses emitted by the AE source are stochastic and characterized by a short duration, on the order of nanoseconds. Since the propagation of the AE signals has a dispersive character, when propagating from the emission point to the AE sensor, the signal duration increases and reaches 100–1000 μs at a distance about 10 meters, and the signal waveform is also determined mainly by the phases of different frequency components of the Lamb modes. In this case, the AE impulse, which is initially stochastic in nature, becomes principally deterministic, and its waveform and spectrum are mainly determined by the parameters of the waveguide.

Based on the above, the following filtering method is applicable: for all located AE events, two $RT$ parameters ($RT_1$ for sensor 1 and $RT_2$ for sensor 2) and two $RA$ parameters ($RA_1$ for sensor 1 and $RA_2$ for sensor 2) are evaluated. Then, the correspondence between the values of the $RA$, $RT$ parameters and a defect distance are determined. If the AE event is located at the distance $x$ from the 1st sensor, the parameters $RT_1$ and $RA_1$ would correspond to the distance $x$, and $RT_2$ and $RA_2$ would correspond to distance ($L–x$), where $L$ is the distance between the sensors and $x$ is the estimated distance between the AE source and the sensor. An AE event is considered as corresponding to the propagation of a defect if, in addition to the standard location criteria associated with the discrepancy of the location

amplitude, *RA* and *RT* parameters estimated for AE impulses measured with both AE channels correspond to the reference value with an error less than 30%.

*RA* and *RT* parameters reference values vs. distance for both measuring channels are shown in Figure 10.

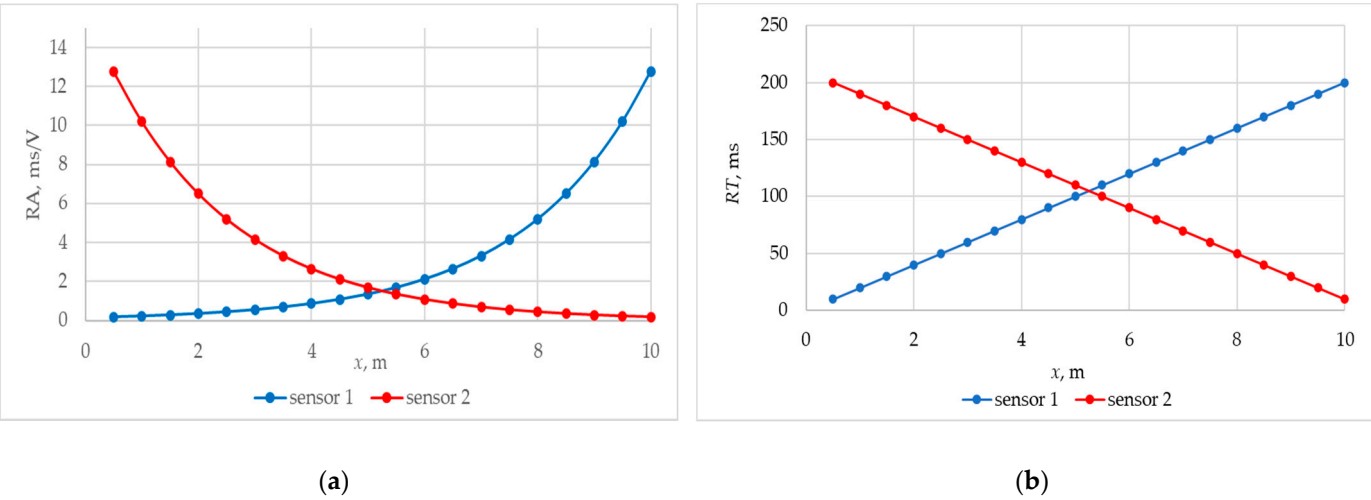

(**a**)                                                                                          (**b**)

**Figure 10.** Parametrical model of AE impulse: (**a**) *RA* depending on the coordinate of AE source, (**b**) *RT* depending on the coordinate of AE source.

## 3. Results

The approaches proposed in this work were tested during structural health monitoring of an ESH 20.90 walking dragline excavator. The reason for the AE testing was a previous damage to the left dragline back legs (a crack in the weld). For data acquisition, the INTERUNIS-IT A-Line 32D DDM system was used, equipped with 30 measuring channels, each of which was connected to resonant sensors (AE GT200) with a bandwidth of 140–200 kHz and a resonance frequency of 180 kHz. The measurement was carried out in the frequency range from 100 to 400 kHz, which ensures a low intensity of the noise environment. AE sensors were installed on the main metal structural dragline components: three sensors for the left and three for the right back legs, eight for the upper boom, four for the lower boom, four for the lower back legs and three for the mast. The distance between the AE sensors varied between 5 and 11 m. Figure 11 shows the location of AE sensors on a dragline boom.

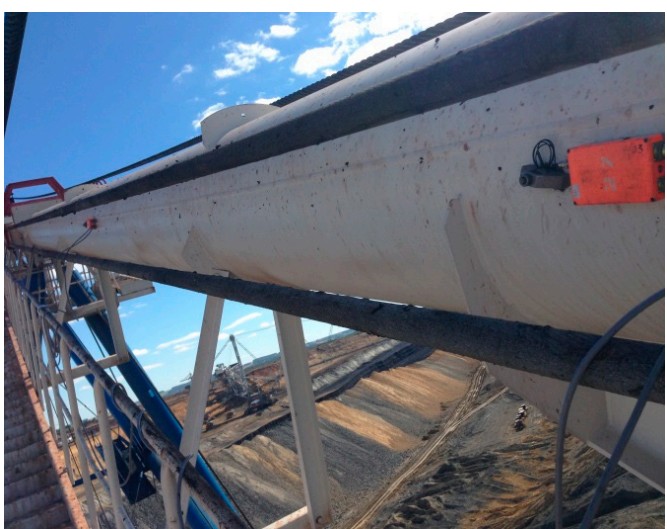

**Figure 11.** Location of AE sensors on a dragline boom.

Intense acoustic noise was measured during dragline operation cycle. The amplitudes of the noise impulses covered almost the entire dynamic range of the AE system from the threshold level of 50 dB to 90 dB. The noise activity reached about 55 AE hits per second on average. Figure 12 shows the values of AE impulse amplitude and cumulative AE hits obtained within an hour. More than 200,000 impulses were registered during an hour of observation. Since the defect emissivity estimated in Section 2.1 was approximately 100 hits per loading cycle, it was obvious that most AE data were caused by noise and this case can hide the presence of a dangerous defect.

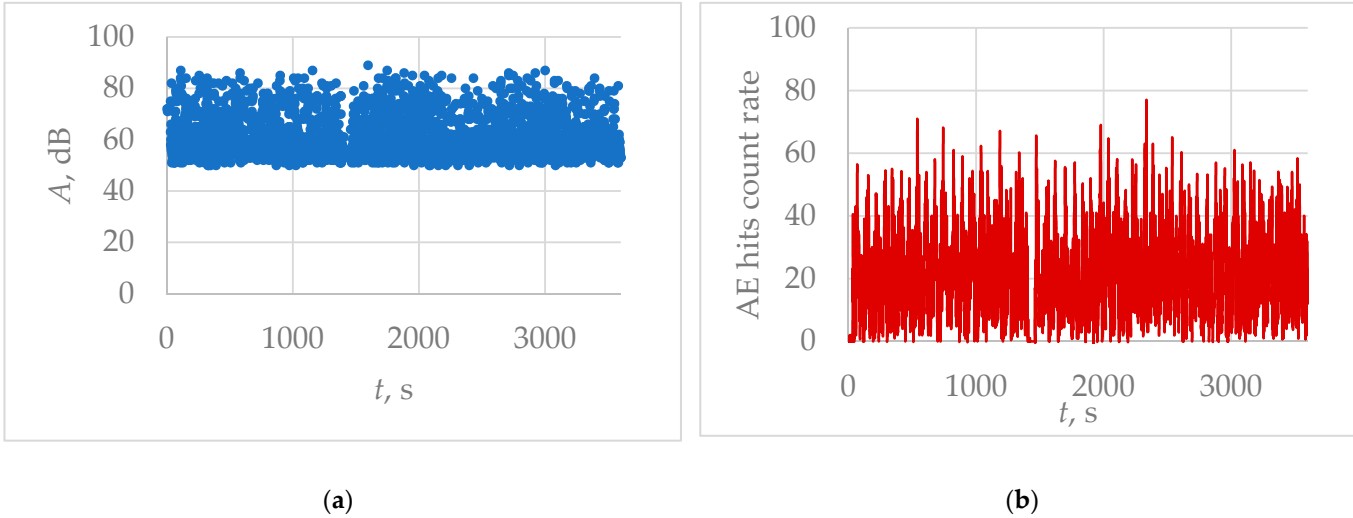

(**a**)                                                                    (**b**)

**Figure 12.** AE data measured during structural health monitoring of walking dragline excavator: (**a**) amplitudes of AE impulses in a period of an hour, (**b**) AE hits count rate in a period of an hour.

The most radical noise filtering result was reached with the help of the location procedure. The main source of noise was the ropes movement, and it did not correspond to a specific spatial coordinate displayed on the "location". Figure 13 presents the linear location diagram of AE sources on the right back leg. The location diagram was formed by two antennas, the first one included sensor 1 and sensors 2, the second one—sensor 2 and sensor 3. AE sensors were located at 1 m, 9 m, and 15 m from the upper edge of the bac leg, respectively. Figure 13 reveals the result of location of AE data measured during monitoring of the dragline in the period of a day. Figure 13a shows a location result without preliminary data filtering; out of 4,000,000 AE hits measured on average by each measuring channel, about 1.5%, of the order of 16,000 pulses, were located. Even though most of the noisy data was discarded, the location diagram displayed much noise due to the uniform filling which was not typical for real AE sources.

Figure 13b depicts the location using the standard criterion for the location amplitude discrepancy. The AE signal pulse amplitude was recalculated taking into account the attenuation coefficient $\alpha$ from the registration point to the location point where the potential defect was located. Moreover, this amplitude did not exceed the permissible range of the AE sensors sensitivity (6 dB). Location amplitude filtration made it possible to exclude about 60% of false locations from the location diagram. The number of localized AE events decreased to 5125, while a localized cluster was formed in the center of the back leg in the welded joint ($x$ = 8500–8800 mm). The increase in the number of indications along the location diagram edges had a noisy nature and it could be explained by friction between the fixing elements of the back leg.

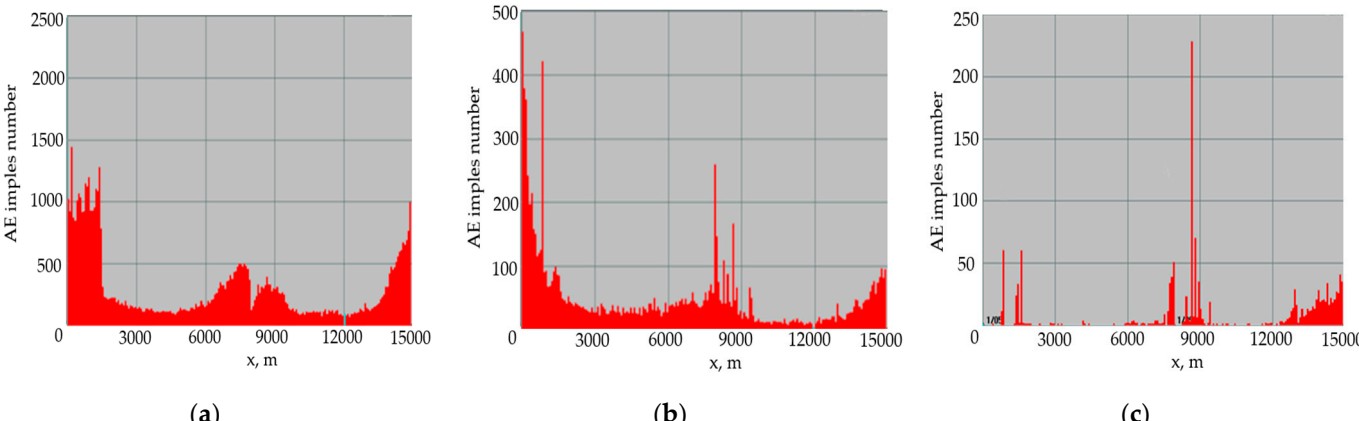

**Figure 13.** Results of AE event location on a right back leg of a dragline: (**a**) without filtering, (**b**) result of the amplitude filtering, (**c**) result of the spatial parametric filtering.

Parametric filtering turned out to be the most considerable approach to noise disposal. The result of its application is represented in Figure 13c. The number of localized AE events decreased to 1517, while both uniform background noise and false clusters at the location diagram edges were removed from the diagram. However, the number of events that in the location cluster in the area of $x \approx 8800$ mm did not change.

To verify the reliability of the obtained results, ultrasonic testing was carried out in the area of the identified location (i.e., in the welded joint of the right back leg). As a result of the control, a crack in the welded joint with a length of 27 mm was revealed. It formed due to the lack of root penetration.

## 4. Discussion

Within the framework of this study, the possibility of the AE structural health monitoring of a dragline excavator was confirmed, and a method for interpreting diagnostic data was developed. This method allows to identify AE signal pulses which are responsible for a defect presence despite the background of intense noise.

The distinctive feature of this study, compared with other approaches to AE monitoring of objects with a high noise level, is that a priori information was widely used in the construction of data processing algorithms. At the preliminary stage of the work, a detailed analysis of all AE monitoring technology components was performed: the AE source parameters, the acoustic path characteristics, and the noise features. The cumulative analysis of a priori information showed that the average noise level was several times higher than the average level of the AE pulse amplitudes. In addition, the intensity of the noise exceeds the AE activity by more than an order of magnitude. This is a significant obstacle for monitoring; however, the analysis of longtime noisy impact showed that its maximum level is about 80 dB. It also has an impulsive nature with quiet periods up to 40–45 dB, which is at least 50% of the time signal realization. Since the phase of the low noise level corresponds to the phase of loading the dragline bucket, it is possible to detect AE signal pulses against the background of noise.

The AE diagnostics is based both on setting the threshold value of the amplitude discrimination below the noise level and on identification of the AE signal pulses by classification of the signal pulses and noise impulse components. The classification is done with the use of an acoustic waveguide parametric model. Classification criteria are specific and allow to remove up to 97% of noise data that differ in parameters from the AE signal pulses.

The main advantage of the proposed approach is that the whole extended structure can be controlled due to the low discrimination threshold. At this time, AE monitoring systems are now commonly used only for the monitoring of the regions with the existing

defects. The reliability of the obtained results is confirmed by the fact that a defect in the weld of the dragline back leg was determined based on the AE monitoring data.

The results obtained in the study have an applied character and a certain practical significance. They are useful for application in the control or monitoring of the dragline metal components during operation.

Within the framework of the research, four dragline excavators "E-Sh-20.90", produced at the "Uralmash" plant from 1985 to 1994, were examined. For each of them, similar parameters of the noise process and close values of the acoustic waveguide parameters were observed. Despite the fact that defects were detected only in one of the monitored objects, the noise filtering procedure was effective in each case under consideration.

## 5. Conclusions

In this paper, a new technique for detecting defects in metal structures of a dragline walking excavator in operating mode is proposed. The technique was implemented using the structural health monitoring system based on the AE method application. The main advantage of the proposed approach is the intelligent processing of diagnostic data based on the results of a preliminary study of defect parameters, acoustic waveguide characteristics, and noise process parameters. Taking into account this information made it possible to design a technique capable of detecting AE pulses corresponding to defects against a background of high-intensity noise. The effectiveness of the proposed approach is confirmed by the detection of a defect in the back leg weld of an "ESH-20.90" dragline excavator.

**Author Contributions:** Conceptualization, V.B. (Vera Barat) and S.E.; methodology, V.B. (Vera Barat) and V.B. (Vladimir Bardakov); software, V.B. (Vladimir Bardakov); validation, V.B. (Vera Barat) and D.K.; formal analysis, S.U.; investigation, V.B. (Vladimir Bardakov) and D.K.; resources, D.K. and S.E.; data curation, V.B. (Vera Barat), M.K. (Marina Karpova), M.K. (Mikhail Kuznetsov), and A.Z.; writing—original draft preparation, V.B. (Vera Barat); writing—review and editing, A.M.; visualization, S.U.; supervision, S.E.; project administration, D.K. and S.E.; funding acquisition, V.B. (Vera Barat). All authors have read and agreed to the published version of the manuscript.

**Funding:** The research was carried out within the framework of the project "Diagnostics of dissimilar welded joints of pearlitic and austenitic steels by the acoustic emission" with the support of a grant from NRU "MPEI" for implementation of scientific research programs "Energy", "Electronics, Radio Engineering and IT", and "Industry 4.0, Technologies for Industry and Robotics" in 2020–2022 (project No. 20/22-0000028/44).

**Institutional Review Board Statement:** Not applicable.

**Informed Consent Statement:** Not applicable.

**Data Availability Statement:** The data presented in this study are available on request from the corresponding author.

**Conflicts of Interest:** The authors declare no conflict of interest.

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
