# Peer review of "Structural Health Monitoring of Walking Dragline Excavator Using Acoustic Emission"

_applsci, doi:10.3390/app11083420_

Round 1

Reviewer 1 Report

Dear authors, I have studied your article in great detail and I have a number of questions and comments on it, namely:
1. Provide data on the number of operated dragline excavators in the world.
2. What machine-building enterprises produce this mining equipment.
3. Provide statistics of dragline boom steel structure failures.
4. Justify the economic feasibility of using acoustic emission in comparison with other diagnostic methods.
5. For a better understanding of this method, it is necessary to present the structural and functional diagram of the monitoring system.
6. The list of references is critically small, it needs to be updated.
7. During monitoring, it is decommissioned, as the production program of the excavator is due to the downtime of the mining machine.
8. Explain whether it is necessary to carry out additional ultrasonic testing after acoustic emission. This is a matter of economic viability.

Author Response

Dear reviewer!

On behalf of the authors team of I want to thank you for the thorough analysis of our paper.

Please, see the attached file with our comments.

Thank you in advance!

Reviewer 2 Report

Expand abstracts to understand what the authors want to do.
Expand the introduction.
Explain what AE exactly what is.
Specify the frequency range.
Explain the measuring equipment that is used.
The measurement technique is explained briefly, exactly how the measurement and acoustic analysis take place?
How the damage risk assessment takes place?

Some authors Iannace et al. (Fault diagnosis for UAV blades using artificial neural network) performed measurements of the possible risk of rupture of rotating organs, the authors can see these papers.

The authors do not explain how the damage assessment technique works and how it is understood from the noise that the damage is close.

Acoustic measurements give information on rotating parts. In this case, it seems that the noise of a crack is acquired under the effect of fatigue. Are there other applications on other cases?

Did the authors study this case only?
The bibliography must be expanded and inserted correctly.
The paper even if it contains interesting information as it was presented cannot be accepted.

Authors need to better describe the measurement and analysis procedure. Why should an acoustic measurement give me an indication of a possible break?

When I perform acoustic measurements in a noisy environment, am I sure what I am measuring?

did the authors perform tests in anechoic chambers?

have authors developed a theoretical model of prediction?

The discussion paragraph must also be separated from that of the conclusions.

Author Response

Dear reviewer!

On behalf of the authors team of I want to thank you for the thorough analysis of our paper.

You indicated in your review that the article requires a major revision. At the same time, in addition to the changes that we make to our article, we also present to your attention additional comments and explanations to your comments. Perhaps, taking into account this information, your view of the article will be changed.

Thank you in advance!

Round 2

Reviewer 2 Report

check References

after ref. 15 the references are incorrect

please correct the references sequence

Author Response

Dear Reviewer! We are deeply grateful to you for working on our article! Perhaps, the reference list became incorrect due to the conversion of the file format before it was posted on the MDPI website. We are sending a revised version. Hope it is OK this time. Thanks again!
